# Affective touch reduces histamine evoked itch experience

**Syed Hasan Ali**[1]*, **Nicholas Fallon**[1], **Timo Giesbrecht**[2], **Andrej Stancak**[1], **Carl A Roberts**[1]

**1** Department of Psychology, University of Liverpool, Liverpool, United Kingdom, **2** Unilever R&D, Port Sunlight, United Kingdom

* Hasan.Ali@liverpool.ac.uk

## Abstract

Itch is a commonly experienced symptom of skin diseases such as eczema. Topical corticosteroid medications are widely used in chronic itch conditions but can lead to skin thinning, and in certain cases, topical corticosteroid withdrawal. As such, non-pharmaceutical alternatives are being researched. The present research explored affective touch (slow stroking, gentle touch signalled by C-tactile afferents) as a strategy to reduce histamine induced itch. Whilst experiencing histamine induced itch on the volar side of the forearms/wrist, participants (n = 60) were subjected to 3 experimental conditions of modulatory somatosensation applied to the volar aspect of the same forearm relative to the site of itch induction (18 trials of each); 1) affective touch (stroking the forearm with a soft brush at 3 cm/s), 2) non-affective touch (stroking the forearm with a soft brush at 18 cm/s) and 3) active control (static brush tapping on the forearm at 1 Hz). Participants were asked to rate the severity of itch, and pleasantness of touch, after each trial. We also investigated whether changes in itch severity scores during the affective touch condition were moderated by individual differences in somatosensory experiences and attitudes as measured on the Touch Experiences and Attitudes Questionnaire (TEAQ), and the Pain Vigilance and Awareness Questionnaire (PVAQ). A linear mixed effects model indicated a main effect of condition on itch severity, whereby affective touch significantly reduced itch severity compared to non-affective touch ($p < .001$) and active control ($p < .001$). The TEAQ and PVAQ scores did not correlate significantly with itch scores in the affective touch condition. These results suggest that affective touch has a relieving effect on histamine-induced itch. Our findings lend further credibility to the idea that affective touch might be able to serve as an effective non-pharmaceutical treatment of itch conditions complementing established approaches.

## Introduction

Itch is defined as an unpleasant sensation leading to the desire to scratch the afflicted area. Chronic itch is defined as itch that lasts longer than 6 weeks [1], and has a lifetime prevalence of 22% [2]. The sensation of itching is a primary complaint in dermatological conditions [3], and is associated with increased rates of depression [4], and reduced quality of life [5,6]. Given the prevalence of chronic itch, and dermatological conditions that cause itch, efforts have been

**Data availability statement:** Anonymised data along with analysis scripts can be found at the University of Liverpool data catalogue (DataCat). The data can be accessed using the following link https://doi.org/10.17638/datacat.liverpool.ac.uk/2767.

**Funding:** This work was supported by a UKRI BBSRC iCASE studentship with industrial partner Unilever (grant number: BB/W51049X/1). TG works for Unilever. Unilever produces beauty and personal care products. However, no Unilever products were used in this study and no data were collected from participants as to their use, or not, of Unilever products.

**Competing interests:** CAR, NF and AS declare that they have received funding from Unilever. TG is an employee of Unilever which markets beauty, personal care and wellbeing products. This does not alter our adherence to PLOS ONE policies on sharing data and materials.

made to understand the neurophysiological basis of itch, in order that more-effective treatments may be developed.

Neurophysiological evidence from microneurography recordings from individual nerve fibres in humans, suggest that itch is primarily mediated by the peripheral afferent system of unmyelinated, slow conducting C-fibres [7]. It is understood that C-fibre afferents convey the affective properties of somatosensory signals such as itch and pain, as opposed to thickly myelinated, rapidly conducting Aβ and Aδ afferents which signal the tactile-discriminatory properties of touch and pain, respectively [8,9]. Microneurography experiments have further expanded the current understanding of peripheral mechanisms of itch and have shown that application of pruritic (itch inducing) histamine preferentially activates a subset of mechanically insensitive histamine sensitive C-fibres (C-Mi) where the time-course of activation of these fibres coincides with the psychophysical sensation of itch [10–12]. However, C-Mi are also stimulated by nociceptive stimuli such as bradykinin and capsaicin [13,14] suggesting that C-Mi fibres are not itch-specific but instead are itch-selective, responding to both pruritic and nociceptive stimuli. This close and complementary relationship between itch and pain is further supported by evidence of individuals with congenital insensitivity to pain also being insensitive to itch [15].

fMRI studies show structural and functional changes associated with chronic itch compared to healthy individuals [16,17]. There is also evidence to show overlaps in brain regions associated with chronic pain and itch, suggesting that chronic pain and itch share central sensitisation mechanisms. For example, in a recently published meta-analysis by Roberts, Giesbrecht [18], they report that itch and pain activate overlapping, but discernible, brain regions such as the thalamus and the insula. Both the thalamus and in insula are activated during itch and pain and are reflective of the emotional-affective components of itch and pain [19,20]. Taken together, this suggests that neural activity, structure, and functioning differs significantly during itch from healthy controls and that there is overlap of brain activity between itch and pain.

Another sensation, albeit distinct perceptually, which is also signalled by C-fibres is affective touch. Affective touch is a slow, gentle caress of touch that is perceived as pleasant and is signalled by another subclass of C-fibre known as the C-tactile (CT) afferent [9]. CTs are located primarily on non-glabrous skin of the arms, legs, and trunk of the body and have a slow conduction velocity of 1 m/s, making them poorly suited to tactile discrimination [8,21]. Studies have shown that CTs exhibit favourable responses to touch stimulation in the 1-10 cm/s range with the most optimal responses being observed at 3 cm/s and a notable inverted-U function being produced when pleasantness ratings are examined in relation to stroking velocity [8,21,22]. In addition to velocity tuning, CTs also demonstrate preference for temperature with stroking at human body temperature resulting in greater responses and pleasantness ratings [23]. During affective touch, activation is seen in the primary somatosensory cortex, secondary somatosensory cortex, prefrontal cortex, orbitofrontal cortex, posterior insula cortex and the anterior cingulate cortex [24–27]. Activation in these areas is associated with the social evaluations, hedonic processing and decision making related to the central processing of affective touch [8,9,28]. In summary, this suggests that CTs constitute a distinct neural pathway for the processing of the positive-affective value of affective touch.

In addition to the distinction made between discriminative and affective touch, there is also an important distinction between the peripheral and central encoding of stimuli which extends to both itch and affective touch. Where peripheral afferents encode for the physical properties of a stimuli, it is central mechanisms, i.e., the brain, that process the subjective percept of these sensations and this aspect of perceptual processing is arguably prone to individual variability. Croy, Bierling [29] demonstrated individual variance in the inverted-U

function between brushing velocity and touch pleasantness previously reported by Loken, Wessberg [22]. Ali, Makdani [30] further expand on this, demonstrating that attitudes to intimate touch, as well as measures of stress, are significant predictors of affective touch pleasantness. Similar outcomes are observed for itch and pain where a greater focus on bodily sensations is associated with higher levels of experienced itch and pain [31]. Taken together, this suggests that individual variance may be able to influence the relieving effect of affective touch on itch.

The skin contains several classes of low-threshold mechanoreceptors (LTMRs) that mediate dissociable aspects of somatosensation. Whilst C-fibre afferents convey emotional properties of itch, and CTs encode pleasurable properties of touch, other LTRMs that are myelinated, thick in diameter and rapidly conducting (e.g. Aβ afferents) send information to the spinal cord about innocuous touch [9,32]. These various somatosensory LTMRs have long been proposed to interact in a gated fashion. For example, Melzack & Wall's (1965) Gate Control Theory suggests a 'gating' mechanism to explain how non-painful somatosensory sensations can reduce pain (i.e., explaining why we rub a painful bump on the head). In this theory a 'gating' process occurs in the substantia gelatinosa (SG) of the spinal cord whereby small diameter afferent fibres (e.g., those that transmit pain and itch and affective touch) 'open' the gate allowing sensory input to the thalamus and beyond, whereas large diameter fibres (e.g., Aβ-fibres) 'close' the gate to inhibit continuing pain (or itch) signals to the brain [33]. However, this theory does not explain how affective touch, which is mediated by small diameter CTs, is able to reduce pain, as seen in previous research [34–37]. One explanation could be that affective touch opens the gateway between somatosensory and affective regions of the brain, resulting in a descending modulatory effect on nociceptive activity [38]. Regardless of the precise mechanisms that underpin this phenomenon, it is entirely possible that this modulatory effect of affective touch on pain may extend to other nociceptive input, e.g., itch.

To date, no studies have investigated whether affective touch can reduce histamine evoked itch severity. Histamine is the most common and earliest described endogenous pruritogen and inflammatory mediator in humans thus, it remains as the gold standard within the field of itch research [39,40]. It exerts its effects via stimulation of H1 receptors on histamine sensitive C-Mi [12]. Histamine has two main routes of activity: endogenous whereby histamine is released by dermal mast cells in the skin via degranulation [41], or exogenous where histamine is introduced into the skin where it is then able to interact with receptors such as H1 receptors. Exogenous application can be done through many ways, however of these methods iontophoresis is the least invasive method. Iontophoresis is a method of transdermal drug and transports molecules (i.e., histamine) across the skins' protective barrier, the stratum corneum via a voltage gradient [42–44]. Histamine iontophoresis induced itch is accompanied by axonal reflex flare activity which is identifiable by a welt and redness in the area where the histamine iontophoresis is conducted [13]. Unlike electrically induced itch, which produces sensations of tapping and pain as well as itch [45,46], histamine produces a 'pure' itching sensation, making it a more viable model of real world itching experience. This suggests that histamine iontophoresis is a reliable and valuable model of assessing whether affective touch can reduce itch.

The idea that affective touch can reduce itch experience was recently assessed by Meijer, Schielen [47] who demonstrated that affective touch reduces electrically induced itch experience. The current study aims to build on the work of Meijer, Schielen [47] to investigate whether the itch modulatory effect of affective touch successfully translates to histamine induced itch. Participants will experience histamine induced itch and be subjected to 3 within-subject experimental conditions of brushing touch: 1) affective touch (stroking the forearm with a soft brush at 3 cm/s), 2) non-affective touch (stroking the forearm at 18 cm/s) and 3) active control (static brush tapping on the forearm at 1 Hz).

Given the overlapping mechanisms between itch, pain and affective touch and the inhibitory relationship between itch and pain as well as pain and affective touch [34–37,48] and that affective touch can reduce electrically evoked itch [47], it suggests that affective touch can reduce histamine evoked itch severity. It is expected that affective touch will reduce histamine evoked itch severity to a greater extent than non-affective touch and active control. Furthermore, given that attitudes towards, and experiences of, affective touch can impact the pleasantness of affective touch, this may impact the relieving effect that affective touch has on itch [49]. Therefore, it is expected that having positive attitudes towards affective touch will be related to a greater relief from itch during affective touch. Previous research suggests that greater vigilance and awareness of bodily sensations will increase the amount of experienced itch [31]. Therefore, it is expected that a greater vigilance and awareness of bodily sensations will result in less relief from itch.

## Materials and methods

### Participants

A total of 63 participants were recruited. All participants were between the ages of 18-40 and had to meet the following criteria: not currently pregnant or lactating, no history of neurological or psychiatric condition, not currently taking neurologically active medication, do not have moles or scars covering large portions of both forearms, do not have a severe allergy condition (e.g., anaphylaxis), are not using topical antihistamine and/or steroidal medication, and do not have an itch or pain condition. In addition to this, participants were required to have not consumed alcohol or taken psychoactive substances (i.e., LSD, MDMA etc) for a minimum of 24 hours prior to attending the lab session. All participants were reminded of their right to withdraw from the study at any point without having to give a reason. Participants were reimbursed £10 for their time or given course credit if they were students on the University of Liverpool Psychology undergraduate degree. Participants were recruited between 21/11/2023 – 05/02/2024. The study was approved by University of Liverpool Central Ethics Committee (Reference number: 10751).

Three participants had to be excluded due to technical issues with saving of data files. A total of 60 participants (5 male, 55 female) were included in the final analysis (mean age = 20.4, standard deviation = 2.61).

### Materials

Touch Experiences and Attitudes Questionnaire (TEAQ): A 57-item questionnaire that aims to measure individuals' experiences of, and attitudes towards, different forms of affective touch [50]. Participants are required to respond to how much they agree with items on the questionnaire, using a 5-point Likert scale, ranging from strongly disagree to strongly agree. The TEAQ consists of 6 subscales: 1) Friends and Family Touch (FFT) e.g. "I find it natural to greet my friends and family with a kiss on the cheek", 2) Current Intimate Touch (CIT) e.g. "I often snuggle up on the sofa with someone", 3) Childhood Touch (ChT) e.g. "There was a lot of physical affection during my childhood", 4) Attitude to Self-Care (ASC) e.g. "I like using body lotions", 5) Attitude to Intimate Touch (AIT) e.g. "I find stroking the hair of a person I am fond of very pleasurable" and 6) Attitude to Unfamiliar Touch (AUT) e.g. "I dislike people being very physically affectionate towards me". A subscale score is obtained through calculation of a mean score per subscale for the items that belong to each subscale. Cronbach's alpha ranges from α=.81 to α=.93 across the 6 subscales, suggesting good internal reliability [50].

Pain Vigilance and Attention Questionnaire (PVAQ) modified: A 16-item questionnaire which was originally developed to assess individual vigilance and attention to pain [51].

The PVAQ contains two subscales, 'attention to pain' (ATP) e.g., "I become preoccupied with pain" and 'attention to changes in pain' (CIP) e.g., "I know immediately when the pain decreases" [51–53]. The PVAQ has been modified to assess bodily sensations as per van Laarhoven, Kraaimaat [31] where the word "pain" is replaced with "bodily sensations" e.g., "I become preoccupied with bodily sensations". We assessed the internal reliability of the new subscales in our sample and found the ATP and CIP subscales to have high internal reliability with McDonald's omega scores of.71 and.69, respectively.

Rating scales for the PVAQ remain the same where participants respond with how frequently they experience bodily sensations on a 5-point Likert scale from 0 – 'never' to 5 – 'always'. A subscale score is obtained through calculation of a mean score per subscale for the items belonging to each subscale.

## Study design

This study used a repeated measures design whereby participants received trials of brushing/tapping (using a 4 cm wide goatshair brush on the volar aspect of the forearm) during 3 blocks of histamine itch induction. Each block of 18 trials consisted of 6 trials from each of the following 3 conditions: 1) affective touch (stroking the forearm with a soft brush at a rate of 3 cm/s), 2), non-affective touch (stroking the forearm with a soft brush at a rate of 18 cm/s) and 3) active control (static brush tapping on the forearm at a rate of 1Hz). Each brushing trial lasted 20 seconds, and trial types/conditions were randomised. A visual metronome visible to the researcher was used to guide the velocity to which the brushing and tapping was performed. This metronome also guided the timing of trials. Participants were brushed on the same arm where the itch was induced.

Following each trial participants were asked to rate their itch severity, and pleasantness of brushing. For itch severity, participants were asked "please rate the current severity of your itch" from "No itch whatsoever" to "Unbearable itch" on a scale from 0 – 10. For brushing pleasantness, participants were asked "please rate the pleasantness of the brushing from "very unpleasant" to "Very pleasant" on a scale from -5 to 5. Participants did not have a time limit in which they had to provide the rating but were instructed to make their ratings as quickly as possible. The rating task was presented using PsychoPy v2022 1.2 [54].

## Stimulation sites

The inside of the wrist and the volar aspects of the forearm were the locations for histamine iontophoresis (see Fig 1 for visualisation of stimulation sites) and brushing respectively. Iontophoresis was conducted approximately 3 cm away from the inside of the elbow, and brushing was conducted on the same arm in a 6 cm long aperture on the forearm 1 cm away from the iontophoresis site. Arms were switched between blocks and the order that participants received stimulation to (e.g., right-left-right, or left-right-left) was counterbalanced.

Histamine Hydrogel: A histamine-methylcellulose hydrogel (2% w/v, 3% w/v respectively) was used to induce itch. The hydrogel was made in house and formulated using histamine chloride (Merck, UK) and methylcellulose powder (Merck, UK) where histamine was diluted using distilled water and pipetted into centrifuge tubes that contained a methylcellulose hydrogel which was prepared beforehand.

## Histamine Iontophoresis and Laser Doppler Perfusion Monitoring

A Moor Instruments Iontophoresis Controller (Moor Instruments, Devon, UK) was used to ionophores the histamine hydrogel into the skin. Iontophoresis was conducted for a total of 60 seconds at 50 µV per block.

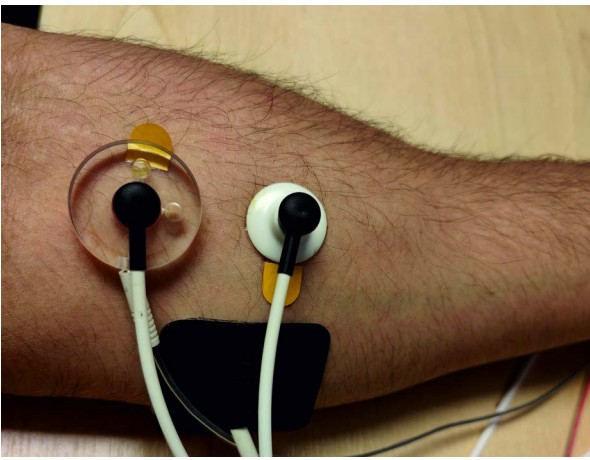

**Fig 1. Iontophoresis set up on arm.** Photo demonstrating the setup of iontophoresis and laser doppler measurement equipment on a participant's arm. The laser doppler probes can be seen on the left where it is placed directly on top of the iontophoresis chamber to measure skin temperature and perfusion through the histamine hydrogel. The right is another doppler probe which is measuring skin temperature and perfusion away from the iontophoresis site. The other electrode for the iontophoresis system can be seen at the bottom of the photograph.

Skin perfusion and temperature was measured using a Moor Instruments Laser Skin Perfusion and Temperature monitor (Moor Instruments, Devon, UK). Two probes were attached to the iontophoresis site, a probe attached directly into the iontophoresis chamber recording wheal activity and another probe attached directly adjacent to the iontophoresis chamber which recorded axonal flare reflex activity (Fig 1). Results from the laser doppler skin perfusion and skin temperature monitors showed that participants demonstrated a standard physiological response to histamine (see supplementary materials for full an explanation of data analysis and results).

## Procedure

Participants were welcomed into the testing room and seated. Participants read the participant information sheet and gave informed consent to participate in the study. Following completion of the consent form, the participants' first forearm was inspected by the researcher for any scars or moles that could interfere with the iontophoresis and laser doppler measurement. If the arm was clear the iontophoresis equipment and laser doppler measuring probe was attached to the arm. If there were tattoos on the forearm, they were avoided to prevent interference with skin perfusion measurement. The iontophoresis chamber was affixed to the skin, and histamine hydrogel was pipetted into the chamber. Following this the laser doppler probes were attached to the skin and iontophoresis chamber. The stimulated arm was then occluded from the participants view using a black cloth sheet on a clothing rail. A 60 second resting baseline taken for the laser doppler skin perfusion and temperature monitoring probes before the iontophoresis was started. After iontophoresis was completed, the itch was allowed to develop for 3 minutes. After this, the experimental trials were initiated. After the first block of trials, equipment was removed from the arm and remaining histamine hydrogel was removed from the arm. Participants then completed the TEAQ and the PVAQ. Once the questionnaires were completed, the same procedure was initiated on the opposite arm for the second block. Once the second block was complete, equipment was removed from the arm and participants were given five minutes to rest before the final block commenced. After completion of the final block, equipment was

removed from the arm and participants were fully debriefed and thanked for their time. Diagram of the study procedure can be seen in Fig 2.

## Data Processing and Analysis

Data was analysed in RStudio [55]. For each participant, average scores for itch severity and brushing pleasantness were calculated for each of the 3 conditions, for each of the 3 bloks of the experiment (i.e., average of two trials per condition for each block), using the 'tidyverse' package in R (version 2.0.0). Examination of histograms and QQ plots of model residuals revealed that the data was normally distributed. As participant ratings for both itch severity and brushing pleasantness were on a continuous scale and our data met the assumptions for parametric analyses, ratings were analysed using a linear mixed-effects model fit using the 'lmer' function from the 'lme4' [56,57] package in R.

For itch severity, a model fit was defined where itch severity was the dependent variable with two fixed effects: brushing condition (affective, non-affective and active control) and block (block one, block two, block three). Participant was included as a random effect in the model.

Brushing pleasantness utilised an identical model fit where pleasantness ratings is the dependent variable with two fixed effects: brushing condition (affective, non-affective and active control) and block (block one, block two, block three). Participant was included as a random effect in the model.

Omnibus effects were tested using Kenward-Roger F tests using the 'ANOVA' function from the 'car' package. Significant effects were followed up using the 'emmeans' function from the 'emmeans' [58] package. Bonferroni corrections were applied to the post-hoc tests for multiple comparisons.

A difference itch severity score was created by subtracting the itch severity during affective touch from itch severity ratings during active control. This created an itch severity change score which was then used to correlate with the subscales of the Pain Vigilance and Attention Questionnaire (PVAQ) and the Touch Experiences and Attitudes Questionnaire (TEAQ). Correlations were also conducted between brushing pleasantness during affective touch and TEAQ and PVAQ subscale scores.

Questionnaires were recoded and scored in SPSS version 28 (IBM, Armonk, NY) according to the authors instructions. The resulting dataset was then imported into R using base R where the subsequent dataset was correlated with the difference itch score that was previously created and brushing pleasantness score during affective touch. The data was correlated with a Pearson's R correlation using the 'rcorr' function from the 'Hmisc' [59] package.

## Results

### Itch severity

A linear mixed effects model showed that there was a significant main effect of *condition* $F(2, 472) = 33.27$, $p < .001$, $\eta p^2 = .12$ and a significant main effect of block on itch severity $F(2, 472) = 54.06$, $p < .001$, $\eta p^2 = .19$ (fig 2). However, there was no significant interaction between condition and block on itch severity. Post-hoc comparisons demonstrated that there was a significant difference in itch severity during affective touch compared to non-affective touch $(p < .001)$ and active control $(p < .001)$. There was also a significant difference in itch severity between active control and non-affective touch $(p < .05)$. See Fig 3 for visualisations.

### Brushing pleasantness

Another linear mixed effects model showed that there was a significant main effect of condition on brushing pleasantness $(F(2, 472) = 353.93$, $p < .001$, $\eta p^2 = .60)$ but no significant main

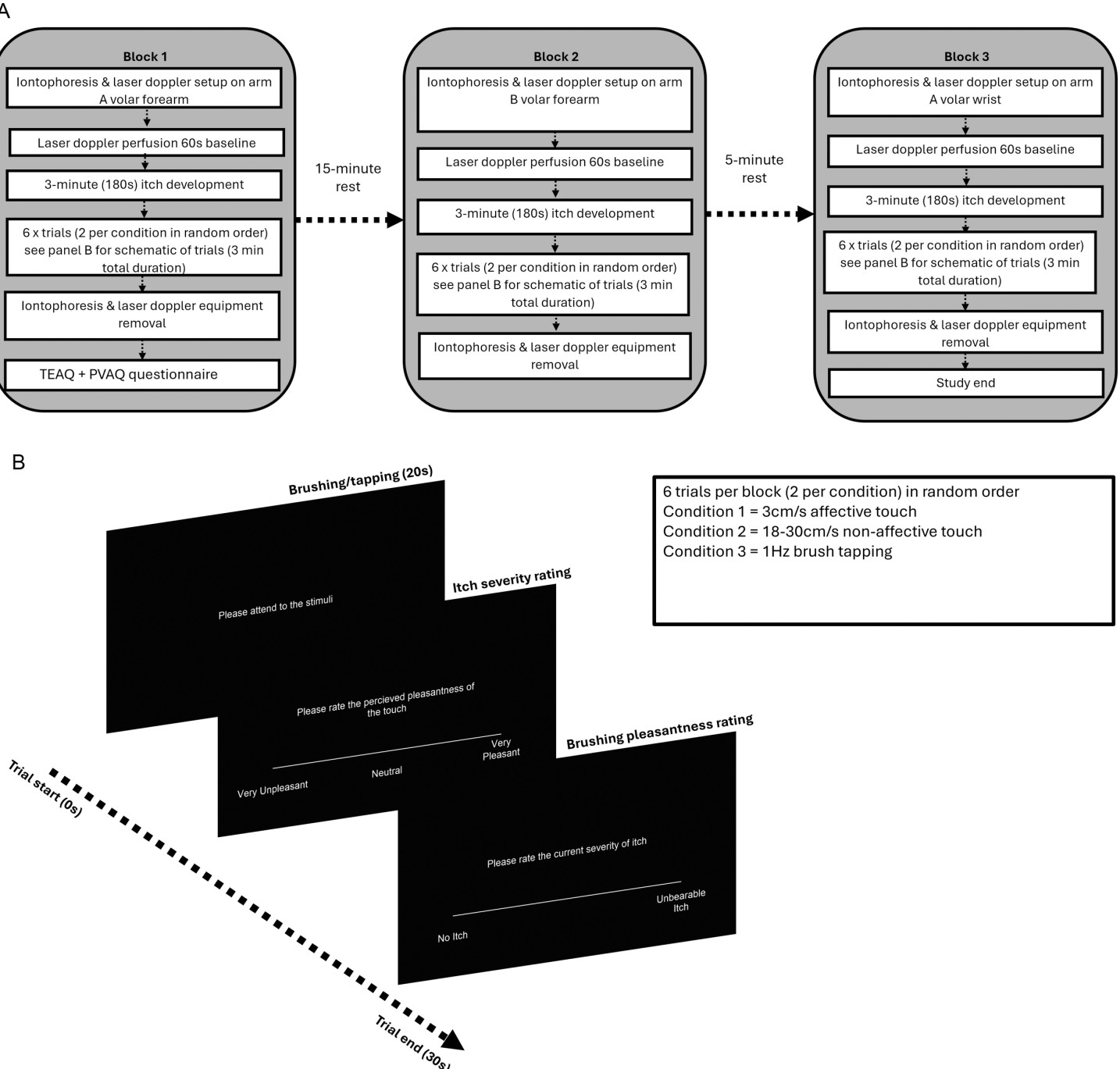

**Fig 2. Schematic diagram of study outline and trials. (A)** Schematic diagram of study procedure showing the order of events for each block. **(B)** Schematic representation of a single trial.

effect of block on brushing pleasantness (fig 3). The interaction between condition and block was not significant. Post hoc comparisons showed that there was a significant difference in brushing pleasantness between active control, non-affective touch *(p < .001)* and affective touch *(p < .001)* and a significant difference between non-affective touch and affective touch *(p < .001)*. See Fig 4 for visualisations.

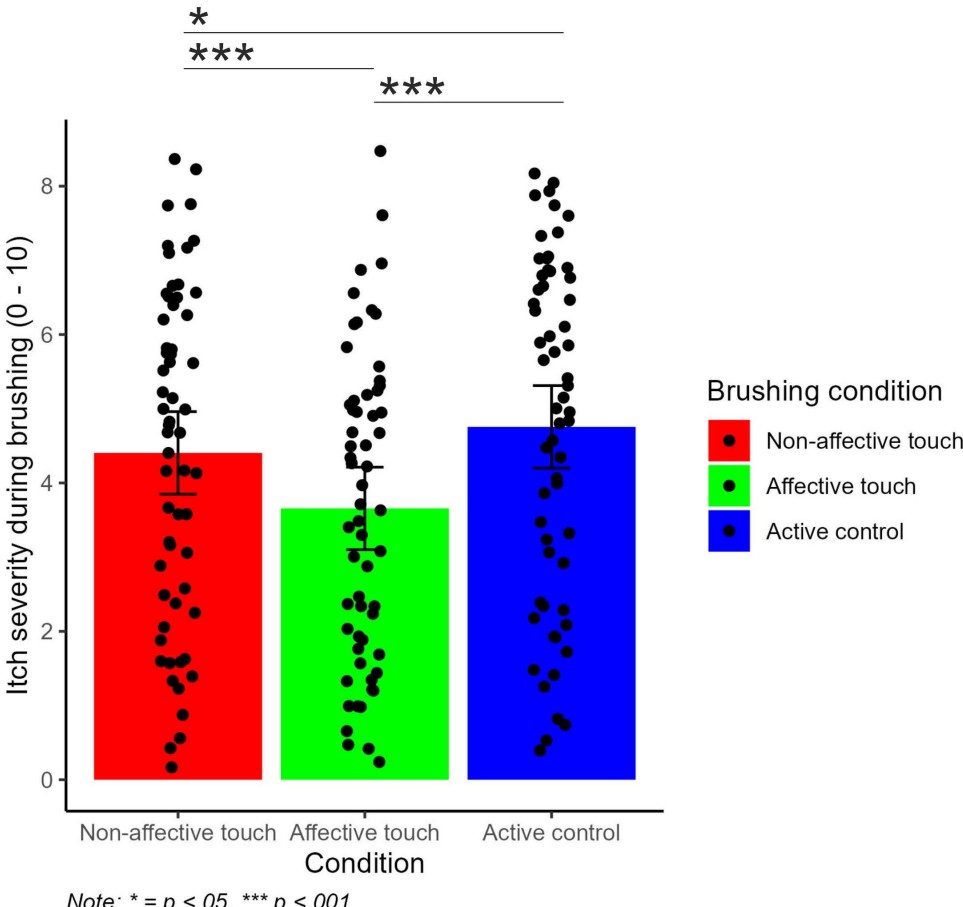

*Note: * = p <.05,  *** p <.001*

**Fig 3. Itch severity by condition.** Affective touch reduced itch severity to a significantly greater extent than non-affective touch and active control brush tapping. Non-affective brushing reduced itch severity to a greater extent than active-control brush tapping.

Further linear mixed effects models showed that there was a significant main effect of condition on brushing pleasantness *(F(2, 472) = 353.93, p <.001, ηp² =.60)* but no significant main effect of block on brushing pleasantness. The interaction between condition and block was non-significant. Post-hoc comparisons showed that non-affective touch was rated significantly more pleasant than active control brush tapping *(p <.001)*. Affective touch was significantly rated to be more pleasant than active control brush tapping *(p <.001)* and non-affective touch *(p <.001)*.

### Relieving effect of touch, pleasantness and PVAQ

There were no significant correlations between difference itch severity scores and any subscale score on the TEAQ or the PVAQ.

There was a significant correlation between the pleasantness of affective touch and the TEAQ AIT subscale *(r = 0.32, p =.01)* Table 1. A partial correction of p ≤ .01 was applied to all correlations to correct for multiple comparisons.

### Discussion

The present research explored affective touch as a strategy to reduce histamine induced itch. Our results showed that affective touch (brushing at 3 cm/s), significantly reduces histamine

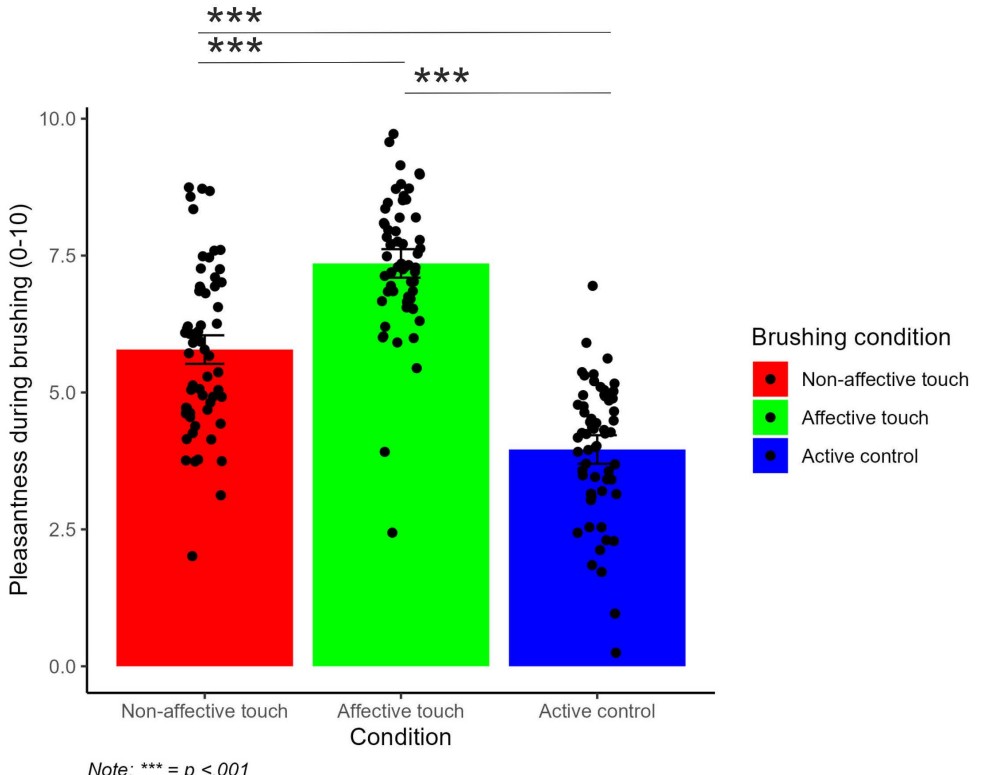

**Fig 4. Mean brushing pleasantness by condition Affective touch was significantly more pleasant than non-affective touch and active control brush tapping.** Non-affective touch was also rated to be significantly more pleasant than active control brush tapping which was rated to be the least pleasant.

**Table 1. Correlation coefficients for brushing pleasantness during affective touch, TEAQ and PVAQ subscale scores.**

| | Difference itch severity | Pleas-antness | PVAQ ATP | PVAQ CTP | TEAQ FFT | TEAQ CIT | TEAQ CHT | TEAQ ASC | TEAQ AIT | TEAQ AUT |
|---|---|---|---|---|---|---|---|---|---|---|
| **Difference itch severity** | – | 0.37 | 0.08 | 0.13 | 0.1 | 0.03 | 0.11 | 0.29 | 0.15 | 0.1 |
| **Pleasantness** | 0.37 | – | 0.05 | 0.02 | 0.14 | 0.06 | 0.06 | 0.22 | 0.32* | 0.22 |

*Note. *p = .01*

induced itch severity compared to non-affective touch (brushing at 30 cm/s), and active control (brush tapping at 1Hz). In addition to this, affective touch brushing was rated as significantly more pleasant than non-affective touch brushing and active control. Scores on the AIT subscale of the TEAQ correlated significantly with pleasantness scores during affective touch brushing. This suggests that positive attitudes to self-care are significantly and positive correlated with pleasantness experienced during affective touch brushing. When correlating subscale scores for the PVAQ with the difference itch severity score during affective touch, no significant correlations were found suggesting that pain vigilance and awareness was not related to the difference in itch severity between affective touch brushing and active control brush tapping.

Our main findings expand on findings by Meijer, Schielen (47) who recently reported reduction in itch severity of an electrically induced itch with affective touch brushing. Taken

together, these data suggest that affective touch can modulate the perceived severity of itch regardless of type of itch induction (electrically evoked or histaminergic). This should provide impetus for further research into these mechanisms in order that non-pharmacological mechanisms for itch conditions may be optimised by focusing on CTs and somatosensory gating.

The precise mechanisms that underpin the relieving effect of pleasant touch on histamine evoked itch experience are yet to be fully elucidated. However, one potential mechanism could be bottom-up inhibition of itch by affective touch in the dorsal horn of the spinal cord. Most of our understanding of such mechanisms is derived from pain research. For example, Gate Control Theory [60] offers a 'gating' mechanism that occurs in the substantia gelatinosa (SG) of the spinal cord whereby small diameter, namely C-fibre afferents 'open' the gate allowing sensory input to the thalamus and beyond leading to the percept of pain and itch, whereas large diameter fibres (e.g., Aβ afferents) 'close' the gate to inhibit continuing pain (or itch) signals to the brain. This theory may explain why non-affective (large diameter Aβ peripheral afferents) touch can reduce itch [61,62]. Recent efforts have been made to understand the specific contribution of small fibre CT mediated affective touch to bottom-up inhibition of pain processing in the spinal cord [34,47,63]. Animal research by Lu and Perl [63] suggests a specific inhibitory pathway related to CT-afferent input in the laminae II of the dorsal horn, whereby CT afferent projections here inhibit nociceptive input in laminae II (and therefore further upstream processing of those signals). It is plausible that our data is reflective of affective touch producing bottom-up inhibition of itch in the spinal cord via both mechanisms due to loading on both Aβ-fibres (i.e., closing the gate) as well as specific effects of CT-afferent input in the laminae II of the dorsal horn.

In addition to bottom-up inhibition of itch signalling, there may also be top-down inhibition at a supraspinal level. For example, somatosensory gating mechanisms in the brain are proposed to prevent the neocortex from being flooded with irrelevant stimuli that could prevent orienting in the physical world [64] and as a result, stimuli that is deemed not to be salient does not reach conscious processing. It is plausible that somatosensory processing is conducted in a hierarchical fashion whereby stimuli of greater salience receive preferential access to the neocortex. It was previously proposed that affective touch is prioritised over itch in the somatosensorial hierarchy due to it having a greater salience and value [65]. Another example of top-down mechanisms is the descending pain modulation system. The descending pain modulation system is a network of cortical and subcortical brain regions that can either facilitate or inhibit nociceptive input via nuclei located in the brainstem [66]. A core aspect of the descending pain modulation system is the periaqueductal grey (PAG) and rostral ventromedial medulla (RVM) axis [67]. The PAG-RVM axis is important in many pro- and anti-nociceptive effects in animals [68]. This axis is governed by the anterior cingulate cortex (ACC), mid-lateral orbitofrontal cortex (OFC) and the ventromedial prefrontal cortex (vmPFC) [69–71]. Coincidentally, these brain areas are also active during affective touch [25,72]. Animal research has corroborated the existence of a descending modulation system by showing the effectiveness of scratching behaviour in reducing pruritic neuronal activity after spinal cord transection [73]. Therefore, it is possible that the descending pain modulation system may have been active during pleasant touch and underpinned the modulation of itch by affective touch. Future neuroimaging studies may be able to determine the central processes that underpin the modulation of itch by affective touch.

An additional avenue through which affective touch might alleviate itch is the neurohormonal pathway. Existing research has pointed to CT fibres as catalysts for oxytocin release [74,75]. Oxytocin, a neurohormone recognized for its influence on positive social behaviours such as pair bonding, has also been linked to stress reduction through low-intensity (affective touch) skin stimulation [76]. Beyond its stress-relieving properties, oxytocin also exhibits

analgesic effects [77], as evidenced in both rats [78] and humans experiencing laser-evoked pain [79]. Given the interconnected nature of itch, pain, and touch, there is a possibility that oxytocin mediates the itch-reducing impact of pleasant touch.

In addition to biological mechanisms, prior research has underscored the involvement of individual variance in attitudes towards and experiences of affective touch in the perception of affective touch as well as awareness of bodily sensations [29–31]. Variations in attitudes towards and experiences of affective touch have been shown to modulate the perceived pleasantness of affective touch in varying contexts [30,50]. Similarly, individual variance in the propensity to direct attention to bodily sensations, such as itch, has been linked to an increased perception of itch severity [31]. Consequently, it was hypothesised that the individual variance may be related to the decrease in itch severity during affective touch brushing. However, no significant correlations were found between itch severity change scores and either subscale of the PVAQ. Similarly, no significant correlations were found between difference itch scores and perceived pleasantness of affective touch in various circumstances. However, there was a significant correlation in pleasantness during affective touch brushing and the Attitude to Self-Care (ASC) subscale in the TEAQ. This implies that neither individual variance in vigilance and awareness of bodily sensation and attitudes towards and experiences of affective touch in specific circumstances were associated with the impact of affective touch brushing on itching severity. This is in contrast to findings from Meijer, Schielen [47] who reported a link between increased awareness of bodily sensations and itch relief through affective touch brushing. This suggests that the main finding of our study is robust as it is less likely to be influenced by individual variances in vigilance and attention to bodily sensations as well as individual variances in the perceived pleasantness of affective touch in various circumstances.

There are limitations with the current study that should be discussed. The method of inducing itch via histamine iontophoresis may induce itch with great variability between individuals. However, this is also an issue experienced with other methods of itch induction such as electrically induced itch, and is accounted for by the within-subjects design. In addition, it is possible that itch intensity was not entirely stable across trials. However, given the randomisation of order of trial type across blocks we do not feel that this impacts the interpretation of the findings.

Whilst histamine iontophoresis is a valuable model in testing whether affective touch can reduce histamine evoked itch, there are other non-histaminergic pathways of itch. For example, the use of the proteinase containing plant spicule cowage that activates the proteinase activated receptor-2 (PAR-2) to produce strong itch 'without flare' typically associated with histamine [20,80–82]. As a result, conclusions drawn from this study do not automatically apply to other types of non-histaminergic itch. Future studies may wish to use models of chronic itch induction such as cowage. Furthermore, the induction of acute itch with histamine iontophoresis may not translate to chronic itch, i.e., eczema.

Another potential limitation of the current study is that histamine was applied to the same arm in blocks one and three, meaning that the first application may influence the response to the third application (e.g., via an additive effect of histamine). However, despite there being a main effect of block for both itch severity and brushing pleasantness, there was no block-by-condition interaction. As such, this does not impact the interpretation of the results.

It is also noteworthy that the majority of participants in the current sample were female, which may reduce the generalisability of the results. Indeed, there is evidence to suggest that females perceive affective touch where females perceive affective touch to be more pleasant than males [83]. However, the results from this study showed that itch severity was reduced independently of the pleasantness of touch. Future studies may wish to recruit equal numbers of males and females to investigate any possible sex differences.

In summary, the present study demonstrated that affective touch reduces histamine evoked itch severity relative to non-affective touch, and active control. These findings contribute to the understanding of the fundamental bioscience of itch and affective touch interactions and lend further credibility to the idea that affective touch might be able to serve as an effective non-pharmaceutical treatment of itch conditions complementing more traditional approaches. Future research may wish to understand the spinal and supraspinal mechanisms underpinning the itch relieving effect of affective touch in order to help translate this research into effective and reliable treatments for itch.

## Supporting Information

**S1 Fig. Laser doppler skin perfusion activity during baseline and during itching.** Compared to baseline, there was a significant difference in the mean wheal and flare response after iontophoresis compared to baseline.
(TIF)

**S2 Fig. Skin temperature during baseline and three blocks of itching and brushing.** Compared to baseline, there was a significant difference in the mean skin temperature wheal and flare response after iontophoresis compared to baseline.
(TIF)

## Author contributions

**Conceptualization:** Syed Hasan Ali, Nicholas Fallon, Timo Giesbrecht, Andrej Stancak, Carl A Roberts.

**Formal analysis:** Syed Hasan Ali.

**Funding acquisition:** Nicholas Fallon, Timo Giesbrecht, Andrej Stancak, Carl A Roberts.

**Investigation:** Syed Hasan Ali.

**Methodology:** Syed Hasan Ali.

**Supervision:** Nicholas Fallon, Timo Giesbrecht, Andrej Stancak, Carl A Roberts.

**Writing – original draft:** Syed Hasan Ali, Carl A Roberts.

**Writing – review & editing:** Syed Hasan Ali, Nicholas Fallon, Timo Giesbrecht, Carl A Roberts.

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
