## [Decision Letter · Decision Letter 0]

15 Nov 2024

<h4>PONE-D-24-28847</h4><h4>Affective touch reduces histamine evoked itch experience</h4><h4>PLOS ONE</h4><h4>

Dear Dr. Ali,

Thank you for submitting your manuscript to PLOS ONE. After careful consideration, we feel that it has merit but does not fully meet PLOS ONE’s publication criteria as it currently stands. Therefore, we invite you to submit a revised version of the manuscript that addresses the points raised during the review process. These concerns are outlined in their reviews which have been included below. We invite you to revise and resubmit the manuscript.

If you will need more time than this to complete your revisions, please reply to this message or contact the journal office at plosone@plos.org . Please include the following items when submitting your revised manuscript:</h4>

<h4>A rebuttal letter that responds to each point raised by the academic editor and reviewer(s). You should upload this letter as a separate file labeled 'Response to Reviewers'.</h4><h4>A marked-up copy of your manuscript that highlights changes made to the original version. You should upload this as a separate file labeled 'Revised Manuscript with Track Changes'.</h4><h4>An unmarked version of your revised paper without tracked changes. You should upload this as a separate file labeled 'Manuscript'.</h4>

<h4>If applicable, we recommend that you deposit your laboratory protocols in protocols.io to enhance the reproducibility of your results. Protocols.io assigns your protocol its own identifier (DOI) so that it can be cited independently in the future. For instructions see: https://journals.plos.org/plosone/s/submission-guidelines#loc-laboratory-protocols . Additionally, PLOS ONE offers an option for publishing peer-reviewed Lab Protocol articles, which describe protocols hosted on protocols.io. Read more information on sharing protocols at https://plos.org/protocols?utm_medium=editorial-email&utm_source=authorletters&utm_campaign=protocols .

We look forward to receiving your revised manuscript.</h4><h4>Kind regards,</h4><h4>Santosh Kumar Mishra

Academic Editor

PLOS ONE</h4><h4>Journal Requirements:

3. Thank you for stating the following financial disclosure: [This work was supported by a UKRI BBSRC studentship with industrial funding from Unilever].

Please state what role the funders took in the study. If the funders had no role, please state: "The funders had no role in study design, data collection and analysis, decision to publish, or preparation of the manuscript.""

4. Thank you for stating the following in the Competing Interests section: [CAR, NF and AS declare that they have received funding from Unilever. TG is an employee of Unilever which markets beauty, personal care and wellbeing products.].

Please confirm that this does not alter your adherence to all PLOS ONE policies on sharing data and materials, by including the following statement: "This does not alter our adherence to PLOS ONE policies on sharing data and materials.” (as detailed online in our guide for authors http://journals.plos.org/plosone/s/competing-interests ). If there are restrictions on sharing of data and/or materials, please state these. Please note that we cannot proceed with consideration of your article until this information has been declared.

Reviewers' comments:

Reviewer's Responses to Questions</h4><h4>**Comments to the Author**

1. Is the manuscript technically sound, and do the data support the conclusions?

The manuscript must describe a technically sound piece of scientific research with data that supports the conclusions. Experiments must have been conducted rigorously, with appropriate controls, replication, and sample sizes. The conclusions must be drawn appropriately based on the data presented.</h4><h4>Reviewer #1: Partly</h4><h4>Reviewer #2: Yes</h4>

<h4>2. Has the statistical analysis been performed appropriately and rigorously?</h4><h4>Reviewer #1: Yes</h4><h4>Reviewer #2: Yes</h4>

<h4>3. Have the authors made all data underlying the findings in their manuscript fully available?

The PLOS Data policy  requires authors to make all data underlying the findings described in their manuscript fully available without restriction, with rare exception (please refer to the Data Availability Statement in the manuscript PDF file). The data should be provided as part of the manuscript or its supporting information, or deposited to a public repository. For example, in addition to summary statistics, the data points behind means, medians and variance measures should be available. If there are restrictions on publicly sharing data—e.g. participant privacy or use of data from a third party—those must be specified.</h4><h4>Reviewer #1: Yes</h4><h4>Reviewer #2: Yes</h4>

<h4>4. Is the manuscript presented in an intelligible fashion and written in standard English?

PLOS ONE does not copyedit accepted manuscripts, so the language in submitted articles must be clear, correct, and unambiguous. Any typographical or grammatical errors should be corrected at revision, so please note any specific errors here.</h4><h4>Reviewer #1: Yes</h4><h4>Reviewer #2: Yes</h4>

<h4>5. Review Comments to the Author

Please use the space provided to explain your answers to the questions above. You may also include additional comments for the author, including concerns about dual publication, research ethics, or publication ethics. (Please upload your review as an attachment if it exceeds 20,000 characters)</h4><h4>Reviewer #1: The authors aimed to examine the effect of affective touch on histamine-induced itch using iontophoresis. While the study presents an interesting observation, there are several concerns that should be addressed:

1) Each block consisted of 6 trials across 3 conditions, and each subject completed 3 blocks. How was the data analyzed to generate a single data point for each condition per subject?

2) Histamine was applied to the same arm in blocks 1 and 3 within a short time span. It's likely that the first application influenced the response to the third application.

3) It seems unlikely that itch intensity remained stable across 18 trials.

4) The study design assumes that the inhibitory effect of affective touch does not carry over to subsequent trials. Is this assumption valid?

5) The complexity of the study design raises questions about its rationale. A schematic of the experimental timeline would be helpful.

6) There is inconsistency between the methods and abstract regarding the locations of histamine iontophoresis and brushing. Was the experiment conducted consistently?

7) Each brushing trial lasted 20 seconds, followed by an itch rating. Were all trial durations consistent?

8) Skin temperature and perfusion results were not reported.

9) Individual data points should be displayed in the graphs.

10) Why were the majority of subjects female?</h4><h4>Reviewer #2: This is an important study that indicates that affective touch can alleviate histaminergic itch in human subjects. Overall the experiments are well designed, the analyses is technically sound, and the knowledge gained can have clinical implications.</h4>

<h4>6. PLOS authors have the option to publish the peer review history of their article (what does this mean? ). If published, this will include your full peer review and any attached files.

**Do you want your identity to be public for this peer review?**  For information about this choice, including consent withdrawal, please see our Privacy Policy .</h4><h4>Reviewer #1: No</h4><h4>Reviewer #2: No</h4>

<h4>

While revising your submission, please upload your figure files to the Preflight Analysis and Conversion Engine (PACE) digital diagnostic tool, https://pacev2.apexcovantage.com/ . PACE helps ensure that figures meet PLOS requirements. To use PACE, you must first register as a user. Registration is free. Then, login and navigate to the UPLOAD tab, where you will find detailed instructions on how to use the tool. If you encounter any issues or have any questions when using PACE, please email PLOS at figures@plos.org . Please note that Supporting Information files do not need this step.</h4><h4> </h4><h4> </h4>

---

## [Author Response · Author response to Decision Letter 1]

6 Jan 2025

Dear Mr Santosh Kimar Mishra,

We would like to thank the reviewers for their comments and the invitation to revise our manuscript titled “Affective touch reduces histamine evoked itch experience” (manuscript number: PONE-D-24-28847). We have addressed each comment below and incorporated changes to the revised manuscript as tracked changes. We feel that the manuscript has now improved significantly and hope this will be acceptable for publication in PlosOne.

Reviewer #1: Each block consisted of 6 trials across 3 conditions, and each subject completed 3 blocks. How was the data analyzed to generate a single data point for each condition per subject?

We thank the reviewer for this comment and agree this needed clarifying in the manuscript. We have added the following to the “Data processing and analysis” section of the manuscript:

“For each participant, average scores for itch severity and itch pleasantness were calculated for each of the 3 conditions, for each of the 3 blocks of the experiment (i.e. average of two trials per condition for each block), using the ‘tidyverse’ package in R (version 2.0.0).”

Reviewer #1: Histamine was applied to the same arm in blocks 1 and 3 within a short time span. It's likely that the first application influenced the response to the third application.

We thank the reviewer for this insight. We have added the following to the discussion to address this point:

“Another potential limitation of the current study is that histamine was applied to the same arm in blocks one and three, meaning that the first application may influence the response to the third application (e.g. via an additive effect of histamine). However, despite there being a main effect of block for both itch severity and brushing pleasantness, there was no block-by-condition interaction, as such this does not impact the interpretation of the results.”

Reviewer #1: It seems unlikely that itch intensity remained stable across 18 trials.

We agree with the reviewer, however we feel that the study design accounted for this with randomisation of trial type within each block. We have added the following to the discussion as a limitation.

“In addition, it is possible that itch intensity was not entirely stable across trials. However, given the randomisation of order of trial type across blocks we do not feel that this impacts the interpretation of the findings.”

Reviewer #1: The study design assumes that the inhibitory effect of affective touch does not carry over to subsequent trials. Is this assumption valid?

Given trials were randomised across blocks, the between conditions differences in effect suggests that inhibitory effects do not carry over to other trial types. Or if they do then actual differences between conditions are in fact even greater.

Reviewer # 1: The complexity of the study design raises questions about its rationale. A schematic of the experimental timeline would be helpful.

We politely disagree with the reviewer about questions over its rationale – especially given that the design of the study has circumvented each of the potential limitations from the 3 previous comments. However, we agree that a schematic would be helpful, and so this has now been added as Fig 2 in the revised submission.

Reviewer #1: There is inconsistency between the methods and abstract regarding the locations of histamine iontophoresis and brushing. Was the experiment conducted consistently?

We must apologise for our error here. The inconsistency between the methods and abstract has now been corrected.

Reviewer #1: Each brushing trial lasted 20 seconds, followed by an itch rating. Were all trial durations consistent?

Each brushing trial was consistently 20 seconds long. This was kept to 20 seconds by a visual metronome visible to the researcher which also guided the speed at which the brushing was done. During the rating of itch severity and brushing pleasantness, participants had as much time as they required but were instructed to make ratings as quickly as possible. The manuscript has now been updated to improve the clarity of the duration of trials.

“Each brushing trial lasted 20 seconds, and trial types/conditions were randomised. A visual metronome visible to the researcher was used to guide the velocity to which the brushing and tapping was performed. This metronome also guided the timing of trials.”

Reviewer #1: Skin temperature and perfusion results were not reported.

We apologise for the confusion here. The skin temperature and perfusion results were conducted to ensure that the participants in our sample had the typical response to histamine iontophoresis. We have added the following to the methods section (below), and added the results to supplementary materials

“Results from the laser doppler skin perfusion and skin temperature monitors showed that participants demonstrated a standard physiological response to histamine (see supplementary materials for full an explanation of data analysis and results).”

Reviewer #1: Individual data points should be displayed in the graphs.

The graphs have now been amended so that individual data points are displayed in the graphs.

Reviewer #1: Why were the majority of subjects female?

This was not intentional but was an effect of opportunistic testing from primarily psychology student sample at the University of Liverpool. We have added the following acknowledgement of this limitation in the discussion:

“It is also noteworthy that the majority of participants in the current sample were female, which may reduce generalisability of the results. Indeed there is evidence to suggest that females perceive affective touch to be more pleasant than males [83]. However, the results from this study showed that itch severity was reduced independently of the pleasantness of touch. Future studies may wish to recruit equal numbers of males and females to investigate any possible sex differences.

Reviewer #2: This is an important study that indicates that affective touch can alleviate histaminergic itch in human subjects. Overall the experiments are well designed, the analyses is technically sound, and the knowledge gained can have clinical implications.

We would like to thank the reviewer for their kind comments and are hoping that the results from this study does have clinical implications.

Yours sincerely,

Hasan Ali

---

## [Decision Letter · Decision Letter 1]

26 Jan 2025

Affective touch reduces histamine evoked itch experience

PONE-D-24-28847R1

Dear Dr. Ali,

We’re pleased to inform you that your manuscript has been judged scientifically suitable for publication and will be formally accepted for publication once it meets all outstanding technical requirements.

Kind regards,

Santosh Kumar Mishra

Academic Editor

PLOS ONE

Additional Editor Comments (optional):

Reviewers' comments:

Reviewer's Responses to Questions

**Comments to the Author**

1. If the authors have adequately addressed your comments raised in a previous round of review and you feel that this manuscript is now acceptable for publication, you may indicate that here to bypass the “Comments to the Author” section, enter your conflict of interest statement in the “Confidential to Editor” section, and submit your "Accept" recommendation.

Reviewer #1: All comments have been addressed

2. Is the manuscript technically sound, and do the data support the conclusions?

Reviewer #1: Yes

3. Has the statistical analysis been performed appropriately and rigorously? 

Reviewer #1: Yes

4. Have the authors made all data underlying the findings in their manuscript fully available?

Reviewer #1: Yes

5. Is the manuscript presented in an intelligible fashion and written in standard English?

Reviewer #1: Yes

6. Review Comments to the Author

Reviewer #1: The authors have adequately addressed the comments I raised in the previous round of review. I have no further concerns.

7. PLOS authors have the option to publish the peer review history of their article (what does this mean? ). If published, this will include your full peer review and any attached files.

**Do you want your identity to be public for this peer review?** For information about this choice, including consent withdrawal, please see our Privacy Policy .

Reviewer #1: No

---

## [Editor Report · Acceptance letter]

PONE-D-24-28847R1

PLOS ONE

Dear Dr. Ali,

I'm pleased to inform you that your manuscript has been deemed suitable for publication in PLOS ONE. Congratulations! Your manuscript is now being handed over to our production team.

Kind regards,

on behalf of

Dr. Santosh Kumar Mishra

Academic Editor

PLOS ONE